# Robot Navigation in Complex Workspaces Employing Harmonic Maps and Adaptive Artificial Potential Fields

**DOI:** 10.3390/s23094464

**Published:** 2023-05-03

**Authors:** Panagiotis Vlantis, Charalampos P. Bechlioulis, Kostas J. Kyriakopoulos

**Affiliations:** 1Department of Electrical and Computer Engineering, University of Patras, 265 04 Patras, Greece; panvlantis@outlook.com; 2School of Mechanical Engineering, National Technical University of Athens, 157 80 Athens, Greece; kkyria@mail.ntua.gr

**Keywords:** motion and path planning, collision avoidance, autonomous vehicle navigation, artificial potential fields

## Abstract

In this work, we address the single robot navigation problem within a planar and arbitrarily connected workspace. In particular, we present an algorithm that transforms any static, compact, planar workspace of arbitrary connectedness and shape to a disk, where the navigation problem can be easily solved. Our solution benefits from the fact that it only requires a fine representation of the workspace boundary (i.e., a set of points), which is easily obtained in practice via SLAM. The proposed transformation, combined with a workspace decomposition strategy that reduces the computational complexity, has been exhaustively tested and has shown excellent performance in complex workspaces. A motion control scheme is also provided for the class of non-holonomic robots with unicycle kinematics, which are commonly used in most industrial applications. Moreover, the tuning of the underlying control parameters is rather straightforward as it affects only the shape of the resulted trajectories and not the critical specifications of collision avoidance and convergence to the goal position. Finally, we validate the efficacy of the proposed navigation strategy via extensive simulations and experimental studies.

## 1. Introduction

The navigation of autonomous robots in cluttered environments is a widely studied topic in the field of robotics. Popular methodologies that have been employed in the related literature to address it include, but are not limited to, *configuration space decomposition approaches* [1,2]; *probabilistic sampling methods* such as rapidly exploring random trees [3,4], probabilistic roadmaps [5,6] and manifold samples [7,8]; and *optimal control strategies* such as receding horizon control [9,10] and path homotopy invariants [11,12]. Apart from the aforementioned discrete methods regarding the workspace and/or the decision domain, Artificial Potential Fields (APFs) that were originally introduced in [13] generally provide a simpler means of encoding collision avoidance specifications, with their negated gradient functioning as a reference motion direction that drives the robot towards the desired goal configuration. As shown in [14], despite their intuitive nature, this class of controllers suffers unavoidably from the presence of unwanted equilibria induced by the workspace’s topology, whose region of attraction may not be trivial. In their seminal work [15], Rimon and Koditschek presented a family of APFs called Navigation Functions (NFs) for point and sphere worlds, as well as a constructive transformation for mapping workspaces cluttered by sequences of star-shaped obstacles into such worlds. However, certain design parameters require tedious tuning to eliminate unwanted local minima and render the transformation a diffeomorphism. In practice, this solution suffers by the fact that the allowable values of the design parameters may cause both the potential and the corresponding transformation to vary too abruptly close to the obstacles (the issue of “disappearing valleys” [15]), thus pushing the trajectories of the robot very close to them. Density functions for remedying such drawbacks or adjustable NFs for relaxing some generally conservative requirements are presented in [16,17]. Additionally, attempts to extend the NF framework directly to non-sphere worlds can be found in [18,19]. Finally, a novel approach based on power diagrams which can be used for designing tune-free vector fields for navigation within convex workspaces is also presented in [20].

Artificial Harmonic Potential Fields (AHPFs) constitute an interesting subclass of APFs, since they are free of unwanted local minima by construction. However, no simple method exists for constructing safe (with respect to obstacle avoidance), harmonic potentials even for simple workspaces. AHPFs suitable for navigation in realistic environments were originally utilized in [21], where computationally expensive numerical techniques were employed to solve the associated Dirichlet and Neumann problems. Several extensions of the aforementioned methodology followed [22,23], addressing issues such as numerical precision and computation, dynamic environments, etc. The panel method was also employed in [24,25,26] to build harmonic potentials to coordinate the motion of single and multiple robots in polygonal environments. In [27,28], well-known closed-form solutions of the uncompressed fluid flow around simple geometries was used in order to safely drive a robot among moving obstacles. Harmonic potential fields have also been used in [29,30] to address the Simultaneous Localization and Mapping problem (SLAM) by coordinating the robot motion in unknown environments. Moreover, a methodology based on the evaluation of the harmonic potential field’s streamlines was used in [31,32] for mapping a multiply connected workspace to a disk, collapsing inner obstacles to line segments or arcs. In a recent work [33], the problem of designing closed form harmonic potentials in sphere worlds was addressed by the introduction of a diffeomorphism [34], which allows mapping such workspaces to the euclidean plane with some of its points removed. Finally, extensions of this work addressing topologically complex three-dimensional workspaces or multi-robot scenarios by introducing appropriate constructive workspace transformations can be found in [35,36], respectively.

### 1.1. Contributions

We address the navigation problem for a robot operating within a static, compact, planar workspace of arbitrary connectedness and shape by designing a control law that safely drives the robot to a given goal position from almost any initial feasible configuration. The goal of this work is twofold. **(A) To cope with the topological complexity of the workspace,** we employed numerical techniques in order to build a transformation that maps the workspace onto a punctured disk and delved into the respective construction in detail. We remark that, although the transformation constructed using this method is an approximation of a harmonic map ideal for navigation, our solution benefits from the fact that it only needs a sufficiently fine polygonal workspace description that can be easily acquired in practice (e.g., through SLAM), contrary to [15,34,36] that require an explicit representation of the workspace boundaries (i.e., as the level sets of sufficiently smooth functions). Moreover, unlike the solutions proposed in [15,36], our approach does not require the decomposition of the workspace obstacles into sequences of simpler overlapping shapes and computes the desired transformation in one step. **(B) To steer the robot to its desired configuration,** we employed a control law based on closed-form AHPFs coupled with adaptive laws for their parameters to eliminate the necessity of explicitly defined local activation neighborhoods around the workspace boundaries for ensuring collision avoidance. Our approach is reactive (closed loop) since it selects the velocity of the robot based on the positions of the robot, the desired goal and the workspace boundary. As such, it is more robust against position measurement errors than other open loop approaches such as configuration space decomposition approaches [2] or probabilistic sampling methods such as rapidly exploring random trees [4], probabilistic roadmaps [6] and manifold samples [8], where an open loop path is initially extracted and executed by a trajectory tracking controller. In this way, even small position errors risk the safe execution of the calculated plan. We remark that our overall control scheme only requires solving a computationally expensive problem once for a given static workspace, independent of the robot’s initial and goal configurations, in contrast to the solutions presented in [21,22]. Finally, we adapt our methodology to the class of differential drive robots, which are commonly encountered in real-world applications and propose an algorithm that decomposes the overall workspace into small neighbouring subsets to render the problem of addressing large workspaces tractable. An overview of the proposed methodology’s pros and cons compared to alternative transformations and potential fields can be seen in Table 1 and Table 2, respectively.

Preliminary results were included in our conference paper [37]. We have to stress though that the algorithmic calculation of the harmonic map is given in the present work, along with a rigorous formulation of the panel method. A modification of the adaptive laws for the parameters of the underlying potential field is also introduced to simplify the tuning process by eliminating the necessity of heuristically defined local activation neighborhoods around the workspace boundaries for ensuring collision avoidance. Moreover, an extension for tackling the navigation problem under unicycle kinematics is also provided. Finally, new comparative simulation results are provided to highlight the strong points of the proposed method with respect to other related works, accompanied by an experiment employing an actual robot navigating within a complex office workspace.

### 1.2. Preliminaries

We use Dr(x) to denote an open disk with radius r>0 centered at x∈R2. Additionally, D and ∂D denote the closed disk and circle with unit radii centered at the origin of R2, respectively. Furthermore, let IN≜{1,2,…,N} and IN⋆≜{0}⋃IN. Given sets A,B⊆Rn, we use clA, ∂A, intA and A¯ to denote the closure, boundary, interior and complement of *A* with respect to Rn, respectively, and A\B to denote the complement of *B* with respect to *A*. Furthermore, we use 0N and 1N to denote the all-zeros and all-ones column vectors of length *N*, respectively, and 0N×M to denote the N×M zero matrix. We also define 1N×Mk, k∈IM as the N×M matrix whose *k*-th column is equal to 1N and every other column is equal to 0N. Given a vector function f(x), we use ∇xf to denote its Jacobian matrix. Furthermore, given an arc *C*, we use |C| to denote its length. We will also say that a set *A* is attractive (repulsive) under a potential function ψ when there exists a point p0∉clA such that if we initialize at p0 and move along the negated gradient of ψ, we will converge (not converge) to ∂A. Finally, a potential function ψ is called harmonic if it satisfies the Laplace equation, i.e., ∇2ψ=0, where ∇2 denotes the Laplacian operator. An important property of harmonic functions is the principle of superposition, which follows from the linearity of the Laplace equation. Moreover, the extrema of a non-constant harmonic function occur on the boundary of the domain of definition, thus excluding any local minima/maxima within it (a desirable property for motion planning).

## 2. Problem Formulation

We consider a robot operating within a compact workspace $W\subset R^{2}$ bounded by a single outer and a finite set of inner disjoint Jordan curves (a Jordan curve is a non-self-intersecting continuous planar closed curve), which correspond to the boundaries of static obstacles. It is assumed that W can be written as:$$W=\bar{W_{0}}\backslash \overset{}{\underset{i\in I_{N}}{⋃}}W_{i}$$
where $W_{i}$, $i\in I_{N}^{⋆}$ denote regions of $R^{2}$ that the robot cannot occupy (see left subplot in Figure 1). Particularly, the complement of $W_{0}$ is considered to be a bounded, simply connected region that may also include a strict subset of its own boundary (this corresponds to cases when we wish to place the robot’s goal configuration on some part of the workspace outer boundary which is not physically occupied by an actual obstacle, e.g., the door of a compartment (refer to Section 5.2 for more details)) and $W_{1}$, $W_{2}$, …, $W_{N}$ are assumed to be closed, simply connected compact sets that are contained in $\bar{W_{0}}$ and are pairwise disjoint. Let $p={\left[x,y\right]}^{T}\in R^{2}$ denote the robot’s position and assume that the robot’s motion is described by the single integrator model:(1)$$\dot{p}=u$$
where $u\in R^{2}$ is the corresponding control input vector.

**Problem 1.** *Our goal is to design a control law to successfully drive a robot with kinematics* (Equation 1) *towards a given goal configuration $p_{d}\in W$ from almost any feasible initial configuration $p_{\mathrm{init}}\in W$, while ensuring collision avoidance, i.e., p(t)∈W for all t≥0.*

**Remark 1.** 
*The results presented in this work can be readily employed for the navigation of disk robots with radius R>0 by appropriately augmenting the workspace boundaries with the robot’s size.*


## 3. Harmonic Maps for Planar Navigation

In this section, we present a methodology that maps the robot’s workspace onto a punctured unit disk, over which the robot’s control law is designed. Particularly, our goal is to construct a transformation, $T:\mathrm{cl}\left(W\right)\mapsto D$, from the closure of the robot’s configuration space $\mathrm{cl}\left(W\right)$ onto the unit disk D with the following properties:T(·) maps the outer boundary $\partial W_{0}$ to the unit circle ∂D;T(·) maps the boundary $\partial W_{i},i\in I_{N}$ of each obstacle to a distinct point $q_{i}={\left[u_{i},v_{i}\right]}^{T}\in \mathrm{int}\left(D\right)$;T(·) is a diffeomorphism for all $p\in \mathrm{int}\left(W\right)$.

To that end, we compute a transformation $\tilde{T}\left(p\right)={\left[\tilde{u}\left(p\right),\tilde{v}\left(p\right)\right]}^{T}$, with $\tilde{u}\left(p\right)$ and $\tilde{v}\left(p\right)$ being harmonic functions with respect to *p*, by approximating an ideal harmonic map *T* that meets the aforementioned properties and the existence of which was proven in Theorem 2 of [38], accompanied by sufficient conditions that render it a diffeomorphism as outlined in the corresponding proof.

Particularly, this theorem implies that given an orientation-preserving, weak homeomorphism $T_{\partial }:\partial W_{0}\to \partial D$ (such a transformation can be easily obtained for any given planar Jordan curve *C* by (1) arbitrarily selecting a point $p_{o}$ on *C*, (2) defining ℓ(p), ∀p∈C as the length of the arc $\hat{p_{o}p}$, assuming one travels from $p_{o}$ to *p* on *C* while having the curve’s interior to its left and (3) choosing $T_{\partial }\left(p\right)={\left[\cos(2\pi \ell (p)/L),\sin(2\pi \ell (p)/L)\right]}^{T}$, where L=|C|) from the workspace outer boundary $\partial W_{0}$ to the boundary of the unit disk, the harmonic map *T* that satisfies the conditions:(2)$$T\left(p\right)=T_{\partial }\left(p\right)≜{\left[\bar{u}\left(p\right),\bar{v}\left(p\right)\right]}^{T},\;\;\;\forall p\in \partial W_{0},$$
(3)$$\int_{\partial W_{i}}\frac{\partial T}{\partial n_{p}}ds=0,\;\;\;\forall i\in I_{N}$$
with $n_{p}$ denoting the unit vector that is normal to the boundary at the point $p\in \partial W_{i}$, $i\in I_{N}$, is a diffeomorphism that maps $\mathrm{cl}\left(W\right)$ to the target set and collapses each inner obstacle $W_{i}$ onto a distinct point $q_{i}$ within its interior (see Figure 1). Note that the coordinates of $q_{i}$, i.e., the images of the internal obstacles, are not explicitly specified but are computed as part of the solution, such that the aforementioned constraints are satisfied.

Given that closed-form solutions to the aforementioned problem are generally not available for non-trivial domains, in this work, we employed numerical techniques and particularly the Panel Method [24,39,40] (similar formulations can be obtained by employing other numerical techniques such as the Boundary Element Method (BEM), the Finite Element Method (FEM) or the Finite Differences Method (FDM)) in order to construct a harmonic map $\tilde{T}$ that sufficiently approximates *T*. As such, by subdividing separately the workspace’s outer and inner boundaries into ${\tilde{M}}_{0},{\tilde{M}}_{1},\ldots ,{\tilde{M}}_{N}$ number of elements (see Figure 2), we define the components of $\tilde{T}\left(p\right)={\left[\tilde{u}\left(p\right),\tilde{v}\left(p\right)\right]}^{T}$ as follows:$$\begin{array}{cc} \; & \tilde{u}\left(p\right)=\sum_{i=0}^{N}\sum_{j=1}^{{\tilde{M}}_{i}}\sum_{l=1}^{{\tilde{M}}_{C}}{\tilde{C}}_{ijl}^{x}{\tilde{H}}_{ijl}\left(p\right)\\ \; & \tilde{v}\left(p\right)=\sum_{i=0}^{N}\sum_{j=1}^{{\tilde{M}}_{i}}\sum_{l=1}^{{\tilde{M}}_{C}}{\tilde{C}}_{ijl}^{y}{\tilde{H}}_{ijl}\left(p\right) \end{array}$$
(4)$${\tilde{H}}_{ijl}\left(p\right)=\int_{{\tilde{E}}_{ij}}{\tilde{G}}_{ijl}\left(s\right)\ln\left(∥p-{\tilde{p}}_{i,j}\left(s\right)∥\right)ds$$
where ${\tilde{M}}_{C}$ is the number of control parameters per element, ${\tilde{E}}_{ij}$ denotes the *j*-th element of the *i*-th boundary’s approximation, ${\tilde{p}}_{i,j}\left(s\right):[0,|{\tilde{E}}_{ij}\left|\right]\mapsto {\tilde{E}}_{ij}$ is a bijective parameterization of ${\tilde{E}}_{ij}$, ${\tilde{G}}_{ijl}:[0,|{\tilde{E}}_{ij}\left|\right]\mapsto R$ is the shape function corresponding to the *l*-th control parameter of ${\tilde{E}}_{ij}$ and ${\tilde{C}}_{ijl}^{x},{\tilde{C}}_{ijl}^{y}\in R$ are control parameters that need to be appropriately selected so that $\tilde{T}$ satisfies properties 1–3 for all $l\in I_{{\tilde{M}}_{C}}$, $j\in I_{{\tilde{M}}_{i}}$ and $i\in I_{N}^{⋆}$. It is worth noting that for common choices of ${\tilde{G}}_{ijl}$ (e.g., constant or linear shape functions) and simple types of ${\tilde{E}}_{ij}$ (e.g., line segments), the integral in (Equation 4) can be easily evaluated to obtain a closed-form expression for ${\tilde{H}}_{ijl}$. As an illustration, for a line segment element ${\tilde{E}}_{ij}$ with two control parameters (i.e., ${\tilde{M}}_{C}=2$), a typical choice for linear shape functions (see Figure 2) is ${\tilde{G}}_{ij1}\left(s\right)=s/\left|{\tilde{E}}_{ij}\right|$, ${\tilde{G}}_{ij2}\left(s\right)=\left(1-s/|{\tilde{E}}_{ij}|\right)$ and ${\tilde{p}}_{i,j}\left(s\right)={\tilde{p}}_{i,j,A}+\left({\tilde{p}}_{i,j,B}-{\tilde{p}}_{i,j,A}\right)s/\left|{\tilde{E}}_{ij}\right|$ for the corresponding parameterization, where ${\tilde{p}}_{i,j,A},{\tilde{p}}_{i,j,B}$ are the element’s end-points. To obtain the unknown control parameters as well as the images of the workspace’s inner obstacles, one needs to solve two independent linear systems of equations:(5)$$\tilde{A}\tilde{X}={\tilde{B}}_{x},\;\;\;\tilde{A}\tilde{Y}={\tilde{B}}_{y}$$
for the unknown vectors:$$\begin{array}{cc} \tilde{X} & ={\left[\begin{array}{c} {\tilde{C}}_{0,1,1}^{x},\cdots ,{\tilde{C}}_{1,1,1}^{x},\cdots ,{\tilde{C}}_{N,{\tilde{M}}_{N},{\tilde{M}}_{C}}^{x},u_{1},\cdots ,u_{N} \end{array}\right]}^{T}\\ \tilde{Y} & ={\left[\begin{array}{c} {\tilde{C}}_{0,1,1}^{y},\cdots ,{\tilde{C}}_{1,1,1}^{y},\cdots ,{\tilde{C}}_{N,{\tilde{M}}_{N},{\tilde{M}}_{C}}^{y},v_{1},\cdots ,v_{N} \end{array}\right]}^{T}. \end{array}$$

The matrix $\tilde{A}$ and the right hand side vectors ${\tilde{B}}_{x}$ and ${\tilde{B}}_{y}$ are constructed by selecting a set of $\sum_{i\in I_{N}^{⋆}}{\tilde{m}}_{i}$ arbitrary points ${\tilde{p}}_{i,j}^{⋆}$ (a typical strategy is to select the points ${\tilde{p}}_{i,j}^{⋆}$ uniformly on the outer and inner boundaries of the given domain) such that (a) ${\tilde{p}}_{i,j}^{⋆}\in \partial W_{i}$ for all $j\in I_{{\tilde{m}}_{i}}$ and $i\in I_{N}^{⋆}$ and (b) $\sum_{i\in I_{N}^{⋆}}{\tilde{m}}_{i}={\tilde{M}}_{C}\sum_{i\in I_{N}^{⋆}}{\tilde{M}}_{i}$, and evaluating (Equation 2) and (Equation 3) at those points as follows:$$\tilde{A}=\left[\begin{array}{cc} {\tilde{A}}_{0}, & 0_{{\tilde{m}}_{0}\times N}\\ {\tilde{A}}_{1}, & -1_{{\tilde{m}}_{1}\times N}^{1}\\ \vdots & \vdots \\ {\tilde{A}}_{N}, & -1_{{\tilde{m}}_{N}\times N}^{N}\\ {\tilde{A}}_{†}, & 0_{N\times N} \end{array}\right],\;{\tilde{B}}_{x}=\left[\begin{array}{c} {\tilde{B}}_{x,0}\\ 0_{{\tilde{m}}_{1}}\\ \vdots \\ 0_{{\tilde{m}}_{N}}\\ 0_{N} \end{array}\right],\;{\tilde{B}}_{y}=\left[\begin{array}{c} {\tilde{B}}_{y,0}\\ 0_{{\tilde{m}}_{1}}\\ \vdots \\ 0_{{\tilde{m}}_{N}}\\ 0_{N} \end{array}\right]$$
$${\tilde{A}}_{k}=\left[\begin{array}{ccc} {\tilde{H}}_{0,1,1}\left({\tilde{p}}_{k,1}^{⋆}\right) & \cdots & {\tilde{H}}_{N,{\tilde{M}}_{N},{\tilde{M}}_{C}}\left({\tilde{p}}_{k,1}^{⋆}\right)\\ {\tilde{H}}_{0,1,1}\left({\tilde{p}}_{k,2}^{⋆}\right) & \cdots & {\tilde{H}}_{N,{\tilde{M}}_{N},{\tilde{M}}_{C}}\left({\tilde{p}}_{k,2}^{⋆}\right)\\ \vdots & \vdots & \vdots \\ {\tilde{H}}_{0,1,1}\left({\tilde{p}}_{k,{\tilde{m}}_{k}}^{⋆}\right) & \cdots & {\tilde{H}}_{N,{\tilde{M}}_{N},{\tilde{M}}_{C}}\left({\tilde{p}}_{k,{\tilde{m}}_{k}}^{⋆}\right) \end{array}\right],\;\forall k\in I_{N}^{⋆}$$
$${\tilde{A}}_{†}=\left[\begin{array}{ccc} \sum_{k=1}^{{\tilde{m}}_{1}}\frac{\partial {\tilde{H}}_{0,1,1}}{\partial n_{0,1}}\left({\tilde{p}}_{1,k}^{⋆}\right) & \ldots & \sum_{k=1}^{{\tilde{m}}_{1}}\frac{\partial {\tilde{H}}_{N,{\tilde{M}}_{N},{\tilde{M}}_{C}}}{\partial n_{N,{\tilde{M}}_{N}}}\left({\tilde{p}}_{1,k}^{⋆}\right)\\ \sum_{k=1}^{{\tilde{m}}_{2}}\frac{\partial {\tilde{H}}_{0,1,1}}{\partial n_{0,1}}\left({\tilde{p}}_{2,k}^{⋆}\right) & \ldots & \sum_{k=1}^{{\tilde{m}}_{2}}\frac{\partial {\tilde{H}}_{N,{\tilde{M}}_{N},{\tilde{M}}_{C}}}{\partial n_{N,{\tilde{M}}_{N}}}\left({\tilde{p}}_{2,k}^{⋆}\right)\\ \vdots & \vdots & \vdots \\ \sum_{k=1}^{{\tilde{m}}_{N}}\frac{\partial {\tilde{H}}_{0,1,1}}{\partial n_{0,1}}\left({\tilde{p}}_{N,k}^{⋆}\right) & \ldots & \sum_{k=1}^{{\tilde{m}}_{N}}\frac{\partial {\tilde{H}}_{N,{\tilde{M}}_{N},{\tilde{M}}_{C}}}{\partial n_{N,{\tilde{M}}_{N}}}\left({\tilde{p}}_{N,k}^{⋆}\right) \end{array}\right]$$
$${\tilde{B}}_{x,0}=\left[\begin{array}{c} \bar{u}\left({\tilde{p}}_{0,1}^{⋆}\right)\\ \bar{u}\left({\tilde{p}}_{0,2}^{⋆}\right)\\ \vdots \\ \bar{u}\left({\tilde{p}}_{0,{\tilde{m}}_{0}}^{⋆}\right) \end{array}\right]\;\;\;{\tilde{B}}_{y,0}=\left[\begin{array}{c} \bar{v}\left({\tilde{p}}_{0,1}^{⋆}\right)\\ \bar{v}\left({\tilde{p}}_{0,2}^{⋆}\right)\\ \vdots \\ \bar{v}\left({\tilde{p}}_{0,{\tilde{m}}_{0}}^{⋆}\right) \end{array}\right].$$

Notice that by discretizing the workspace boundaries into a large number of sufficiently small elements, the overall approximation error between the solution $\tilde{T}$ of the aforementioned linear problem and the exact transformation *T* can be rendered arbitrarily small (see [39,40]). However, the complexity of constructing the mapping is of order $O({\bar{M}}^{3})$, where $\bar{M}$ denotes the number of total elements of the mapping (i.e., the complexity of the solution of the dense system of linear Equation (Equation 5)). Nevertheless, the construction of the transformation, which is the main computational bottleneck, is performed only once at the beginning. Additionally, apart from the straightforward user-defined homeomorphism $T_{\partial }$ on the workspace boundary, no tedious trial and error tuning is needed to extract the diffeomorphic transformation $\tilde{T}$, in contrast to other related works such as the Star-to-Sphere Transformation (SST) [15], the Multi-Agent Navigation Transformation (MANT) [36] and the Navigation Transformation (NT) [34].

## 4. Control Design

To address Problem 1, we equip the robot with the aforementioned transformation q=T(p) from the closure of its configuration space W onto the unit disk D and an artificial potential ψ(q,k) augmented with an adaptive control law $\dot{k}=f_{k}\left(q,k\right)$ for its parameters $k={\left[k_{d},k_{1},k_{2},\ldots ,k_{N}\right]}^{T}$. The robot velocity control law is calculated as follows:(6)$$u=-K_{u}s\left(q,k\right)J^{-1}\left(p\right)\nabla_{q}\psi \left(q,k\right)$$
where J(p) denotes the Jacobian matrix of T(p), s(q,k)≥0 is a continuously differentiable gain function given by:(7)$$s\left(q,k\right)=\gamma \sigma_{p}\left(\frac{1-∥q∥}{\epsilon_{p}}\right)+\left(1-\gamma \right)\sigma_{v}\left(\frac{{\left(\nabla_{q}\psi \right)}^{T}q}{\epsilon_{v}+∥\nabla_{q}\psi ∥∥q∥}\right)$$
with
$$\sigma_{p}\left(x\right)=\left\{\begin{array}{cc} x^{2}\left(3-2x\right), & \;\mathrm{if}\;x\leq 1\\ 1, & \;\mathrm{if}\;x>1 \end{array}\right.,$$
(8)$$\sigma_{v}\left(x\right)=\left\{\begin{array}{cc} x^{2}, & \;\mathrm{if}\;x\geq 0\\ 0, & \;\mathrm{if}\;x<0 \end{array}\right.$$
and $K_{u}$, γ, $\epsilon_{p}$ and $\epsilon_{v}$ are scalar constants such that $K_{u},\epsilon_{v}>0$ and $\gamma ,\epsilon_{p}\in \left(0,1\right)$. More specifically, s(q,k) consists of two individual terms, with the first vanishing as the robot approaches the workspace’s outer boundary (and its distance from the unit circle is less than $\epsilon_{p}$) and the second vanishing when the robot’s velocity points away from the disk’s center. The scalar parameter γ can be used for adjusting the contribution of each respective term of s(q,k). Finally, ψ is a harmonic artificial potential field defined on the image T(W) of the workspace W and whose negated gradient $-\nabla_{q}\psi \left(q,k\right)$ defines the direction of the robot’s motion in the real workspace W via the inverse Jacobian $J^{-1}\left(p\right)$. By design, the resultant vector field precludes collisions between the robot and the workspace’s inner obstacles and renders the goal configuration almost globally attractive except for a set of measure zero initial configurations. However, since $W_{0}$ may not be repulsive under ψ for an arbitrary, fixed selection of *k*, we also introduce the adaptive law $f_{k}\left(q,k\right)$ which, along with s(q,k), guarantees forward invariance of the workspace without compromising the convergence and stability properties of the overall system. The following subsections elaborate on each component of the proposed control law individually.

### 4.1. Artificial Harmonic Potential Fields

We construct an artificial harmonic potential field on the disk space D employing point sources placed at the desired configuration $q_{d}=T\left(p_{d}\right)$ as well as at the points $q_{i}=T\left(\partial W_{i}\right),\;\forall i\in I_{N}$ that correspond to the inner obstacles, as follows:(9)$$\phi \left(q,k\right)=k_{d}\ln\left(\frac{∥q-q_{d}∥}{2}\right)-\sum_{i=1}^{N}k_{i}\ln\left(\frac{∥q-q_{i}∥}{2}\right)$$
where $k_{d}>0$ and $k_{i}\geq 0$ denote harmonic source strengths which vary according to adaptive laws that are presented later. An interesting property of the above potential field, which stems from the maximum principle for harmonic functions, is that, for fixed *k*, the only minima of ϕ are located at $q_{d}$ and, possibly, at infinity. As a direct consequence of this property, the Hessian $\nabla_{q}^{2}\phi$ computed at a non-degenerate critical point of ϕ in our domain’s interior has one positive and one negative eigenvalue with the same magnitude, e.g., λ and −λ with λ>0.

Next, we define a reference potential ψ based on ϕ, which is given by:(10)$$\psi \left(q,k\right)=\frac{1+\tanh(\phi \left(q,k\right)/w_{\phi })}{2}$$
where $w_{\phi }$ is a positive scaling constant. Note that ψ maps the extended real line to the closed interval [0,1]. As $\tanh\left(\phi /w_{\phi }\right)$ is a strictly increasing function, the only critical points of ψ are the ones inherited from ϕ with their indices preserved. Furthermore, the gradient of ψ with respect to *q*, given by
(11)$$\nabla_{q}\psi =\frac{1-{\left(\tanh\left(\phi /w_{\phi }\right)\right)}^{2}}{2w_{\phi }}\nabla_{q}\phi ,$$
is well defined and bounded for all q∈D.

If the workspace was radially unbounded, selecting *k* fixed with $k_{d}>\sum_{i=1}^{N}k_{i}$ would render the potential field (Equation 10) sufficient for navigation. The author of [33] addresses bounded workspaces that are diffeomorphic to sphere worlds by simply mapping the outer bounding circle to infinity. In this work, we would like to be able to place $q_{d}$ on regions of ∂D that are not physically occupied by obstacles (such as passages to other compartments, see, for example, Section 5.2); thus, we cannot follow the same procedure since that would render the effect of the sole attractor on the robot null. Instead, we design appropriate adaptive laws for the parameters *k* of ϕ to render the outer boundary repulsive and establish the forward completeness of the proposed scheme at all times.

Before proceeding with the definition of the adaptive law, we first state two propositions that will be used in the subsequent analysis, the proofs of which can be found in the Appendix A.

**Proposition 1.** 
*Let $k_{d}>0$ and $q^{\prime }\in \partial D\backslash \left\{q_{d}\right\}$. There exists $k^{\prime }>0$ such that if $k_{i}<k^{\prime }$, $\forall i\in I_{N}$, then $q^{\prime }$ is repulsive under ψ.*


**Proposition 2.** 
*If $k_{i}$ are non-negative and bounded, there exists $k_{d}^{\prime }>0$ such that ψ is Morse for all $k_{d}\geq k_{d}^{\prime }$.*


### 4.2. Adaptive Laws

We now present the adaptive law $\dot{k}=f_{k}\left(q,k\right)$ that updates the parameters of the potential field ψ. Its primary goal is to render (a) the workspace outer boundary repulsive and (b) any critical point of ϕ in the vicinity of the robot non-degenerate, a property that will be used later in the analysis. In particular, we consider $f_{k}$ of the form:(12)$$\begin{array}{ll} {\dot{k}}_{d} & =\xi_{1}\left(\lambda +∥\nabla_{q}\phi ∥;\epsilon_{1}\right)\\ {\dot{k}}_{i} & =\left({\bar{k}}_{i}-k_{i}\right)w_{i}\ell_{i}g_{i}-K_{k}k_{i}h_{i}w_{0}\left(g_{0}+\xi_{1}\left(s;\epsilon_{2}\right)\right),\;\forall i\in I_{N} \end{array}$$
where $w_{i}$ and $g_{i}$, $i\in I_{N}^{⋆}$, as well as $h_{i}$, $i\in I_{N}$, are functions to be defined later, ${\bar{k}}_{i}$, $i\in I_{N}$ are desired upper bounds for $k_{i}$, λ denotes the non-negative eigenvalue of $\nabla_{q}^{2}\phi$, $K_{k}$ is a positive control gain and $\epsilon_{1}$ and $\epsilon_{2}$ are small positive constants. The continuously differentiable switch $\xi_{1}\left(x;\epsilon \right)$ and functions $\ell_{i}\left(q\right)$ are, respectively, given by:$$\xi_{1}\left(x;\epsilon \right)=1-\sigma_{p}\left(x/\epsilon \right)$$
(13)$$\ell_{i}\left(q\right)=-\frac{K_{u}s\left(q,k\right)}{\ln\left(\frac{∥q-q_{i}∥}{2}\right)}.$$

According to Proposition 1, our first requirement can be accomplished by designing $f_{k}$ to reduce $k_{i}$ as the robot approaches ∂D. To do so without compromising the inherent inner obstacle collision avoidance properties of ϕ, we need to also ensure that each $k_{i}$ does not vanish within some neighborhood of $q_{i}$ for all $i\in I_{N}$. To that end, firstly we define $g_{i}$, employing the smoothly vanishing function defined in (Equation 8) to serve as pseudo-metrics of the alignment between the robot’s velocity and the directions towards the goal and inner obstacles, respectively, given by:(14)$$\begin{array}{c} g_{i}=\sigma_{v}\left({\bar{g}}_{i}\right),\;\;\;\forall i\in I_{N}^{⋆} \end{array}$$
with
$$\begin{array}{cc} {\bar{g}}_{0} & =\frac{1}{4}\left(\alpha ∥\nabla_{q}\psi ∥∥q-q_{d}∥-{\left(\nabla_{q}\psi \right)}^{T}\left(q-q_{d}\right)\right)\\ {\bar{g}}_{i} & =\frac{1}{2}{\left(\nabla_{q}\psi \right)}^{T}\left(q-q_{i}\right),\;\;\;\forall i\in I_{N} \end{array}$$
where α∈(0,1] is a fixed constant that is used for selecting the desired alignment between the robot’s motion and the direction to the goal. We also define the accompanying weights $w_{i}$ as follows to ensure that only one term of (Equation 12) dominates as the robot approaches a particular boundary of W:(15)$$\begin{array}{cc} w_{0} & =\frac{\xi_{2}\left({\bar{w}}_{0};\epsilon_{3}\right)}{{\bar{w}}_{0}+\sum_{j=1}^{N}\left({\bar{k}}_{j}{\bar{w}}_{j}\right)}\\ w_{i} & =\frac{{\bar{w}}_{i}}{{\bar{w}}_{0}+\sum_{j=1}^{N}\left({\bar{k}}_{j}{\bar{w}}_{j}\right)},\;\;\;\forall i\in I_{N} \end{array}$$
with
$${\bar{w}}_{i}=\bar{r_{i}}/\left(r_{i}+\bar{r_{i}}\right),\;\;\;\forall i\in I_{N}^{⋆},$$
$$r_{0}={\left(1-∥q∥\right)}^{2},\;\;\;\;\;r_{i}={∥q-q_{i}∥}^{2},\;\forall i\in I_{N}$$
(16)$$\begin{array}{c} \bar{r_{i}}=\sqrt[{m}]{\sum_{j\neq i}{\left(r_{j}\right)}^{m}},\;\;\;\forall i\in I_{N} \end{array}$$
$$\xi_{2}\left(x;\epsilon \right)=\left\{\begin{array}{cc} 0, & \;\mathrm{if}\;x<\epsilon \\ {\left(\frac{x-\epsilon }{1-\epsilon }\right)}^{2}\left(3-2\frac{x-\epsilon }{1-\epsilon }\right), & \;\mathrm{if}\;\epsilon \leq x\leq 1\\ 1, & \;\mathrm{otherwise}\; \end{array}\right.$$
for a scalar constant $\epsilon_{3}\in \left(0,1\right)$ in (Equation 15) and some integer m<−1 in (Equation 16) that serves as a smooth under-approximation of $\min_{j\neq i}\left(r_{j}\right),i\in I_{N}$. Finally, the weights $h_{i}$, $i\in I_{N}$ are defined as follows:$$h_{i}=1+\frac{\sigma_{v}\left({\bar{h}}_{i}\right)}{1+\sum_{j\in I_{N}}\sigma_{v}\left({\bar{h}}_{j}\right)}$$
with
$${\bar{h}}_{i}=k_{i}{\left(1-{\left(\tanh\left(\phi /w_{\phi }\right)\right)}^{2}\right)}^{2}{\left(\frac{q_{d}-q}{{∥q_{d}-q∥}^{2}}\right)}^{T}\frac{q_{i}-q}{{∥q_{i}-q∥}^{2}}$$
whose purpose is to accelerate the decay of those $k_{i}$ that contribute the most to the component of $\nabla_{q}\psi$ that pushes the robot toward the workspace’s outer boundary.

Regarding the second requirement, as shown in Proposition 1, selecting a $k_{d}$ above a certain threshold is sufficient to render ϕ free of degenerate equilibria. On the other hand, for a given ${\bar{k}}_{i}$, increasing $k_{d}$ steers the robot closer to the workspace’s inner obstacles. Nevertheless, since the robot may never actually enter the vicinity of a degenerate equilibrium, instead of setting $k_{d}$ sufficiently large a priori, the adaptive law for the parameter $k_{d}$ is introduced to increase $k_{d}$ only when it is actually needed, thus alleviating the aforementioned shortcoming.

### 4.3. Stability Analysis

Let us consider the overall system:(17)$$\begin{array}{c} \dot{z}=f_{z}\left(z\right) \end{array}$$
where z=(q,k) and $f_{z}\left(z\right)=\left(f_{q},f_{k}\right)$ with $f_{q}=Ju$. Furthermore, let Ω denote the image of W under *T*, i.e., Ω=T(W). Note that Ω consists of $\mathrm{int}\left(D\right)$, possibly with a subset of ∂D, with the points $q_{i}$ removed. In this section, we elaborate on the stability properties of (Equation 17) under the proposed control scheme (Equation 6) and (Equation 12). First, we formalize the safety properties of the closed-loop system dynamics, which guarantee that our robot does not collide with any obstacle.

**Proposition 3.** *The workspace W is invariant under the dynamics* (Equation 17) *with control laws* (Equation 6) *and* (Equation 12), *i.e., p(t)∈W for all t≥0.*

**Proof.** For the proof, refer to the Appendix A.    □

Having eliminated the possibility of the robot colliding with the workspace’s boundaries, we proceed by showing that all critical points of ψ, where (Equation 17) may converge to, are either non-degenerate saddles or $q_{d}$. Additionally, we show that the latter is a stable equilibrium.

**Proposition 4.** *The artificial potential ψ decreases along the trajectories of the closed-loop system and its time derivative vanishes only at its critical points. Additionally, the preimage of $q_{d}$ is a set of stable equilibria of* (Equation 1).

**Proof.** For the proof, refer to the Appendix A.    □

**Proposition 5.** 
*Let $z^{⋆}=\left(q^{⋆},k^{⋆}\right)$ be a critical point of the closed-loop system dynamics with $q^{⋆}\in \Omega \backslash \left\{q_{d}\right\}$. Then, $q^{⋆}$ is a non-degenerate saddle point of ψ.*


**Proof.** For the proof, refer to the Appendix A.    □

Finally, we conclude this section with the main theoretical findings.

**Theorem 1.** *System* (Equation 1) *under the control law* (Equation 6) *and* (Equation 12) *converges safely to $p_{d}$, for almost all initial configurations, thus addressing successfully Problem 1.*

**Proof.** For the proof, refer to the Appendix A.    □

**Remark 2.** *Owing to the adaptive laws* (Equation 12) *that modify the harmonic source strengths online to secure the safety and convergence properties at all times, the selection of the fixed control parameters in the proposed scheme, i.e., $K_{u}$, γ, $\epsilon_{p}$, $\epsilon_{v}$, $w_{\phi }$, $K_{k}$, $\epsilon_{1}$, $\epsilon_{2}$, α and $\epsilon_{3}$, is straightforward as it affects only the trajectory evolution within the workspace and not the aforementioned critical properties. Consequently, their values should be set freely as opposed to NFs, where the selection of the main parameters severely affects the convergence properties of the adopted scheme and cannot be conducted constructively for generic workspaces of arbitrary topology.*

## 5. Extensions

In this section, we present certain extensions of the proposed approach to (a) address the safe navigation problem for unicycle robots which are frequently encountered in many application domains and (b) tackle computational complexity issues that affect the numerical computation of the harmonic map presented in Section 3 as the size of the workspace increases.

### 5.1. Unicycle Robot Kinematics

In this subsection, we consider robots whose motion is subjected to Pfaffian constraints of the form:(18)$$\begin{array}{cc} \dot{p} & =n_{\theta }v\\ \dot{\theta } & =\omega \end{array}$$
where θ∈[0,2π) denotes the robot’s orientation, $n_{\theta }={\left[\cos(\theta ),\sin(\theta )\right]}^{T}$, and v,ω∈R are control inputs corresponding to the robot’s linear and angular velocities, respectively. First, let us define the robot’s kinematics in the image of the configuration space via the proposed transformation as follows:(19)$$\begin{array}{cc} \dot{q} & =n_{\hat{\theta }}\hat{v}\\ \dot{\hat{\theta }} & =\hat{\omega } \end{array}.$$

Note that the orientations θ and $\hat{\theta }$ are related by:$$n_{\hat{\theta }}=\frac{J_{p}n_{\theta }}{∥J_{p}n_{\theta }∥}$$
where $J_{p}=J\left(p\right)$. To safely drive the robot to its goal configuration, we consider the following control laws:(20)$$\begin{array}{cc} \hat{v} & =-K_{v}s_{v}\left(q,\hat{\theta },k\right){\left(n_{\hat{\theta }}\right)}^{T}\nabla_{q}\psi \left(q,k\right)\\ \hat{\omega } & =-K_{\omega }{\left(n_{\hat{\theta }}^{⊥}\right)}^{T}\nabla_{q}\psi \left(q,k\right) \end{array}$$
with $K_{v},K_{\omega }\in R$ positive constant gains, $n_{\theta }^{⊥}={\left[-\sin(\theta ),\cos(\theta )\right]}^{T}$ and
$$s_{v}\left(q,\hat{\theta },k\right)=\gamma \sigma_{p}\left(\frac{1-∥q∥}{\epsilon_{p}}\right)+\left(1-\gamma \right)\sigma_{v}\left(\frac{\left({n_{\hat{\theta }}}^{T}\nabla_{q}\psi \right){n_{\hat{\theta }}}^{T}q}{\epsilon_{v}+\left|{n_{\hat{\theta }}}^{T}\nabla_{q}\psi \right|∥q∥}\right).$$

Additionally, we need to employ a modified version of the adaptive law for the potential field parameters, which is obtained by substituting *s* with $s_{v}$ in (Equation 12) and (Equation 13) and ${\bar{g}}_{i}$, $i\in I_{N}^{⋆}$, with
$$\begin{array}{cc} {\bar{g}}_{v,0} & =\frac{1}{4}\left(\alpha \left|{n_{\hat{\theta }}}^{T}\nabla_{q}\psi \right|∥q-q_{d}∥-\left({n_{\hat{\theta }}}^{T}\nabla_{q}\psi \right){n_{\hat{\theta }}}^{T}\left(q-q_{d}\right)\right)\\ {\bar{g}}_{v,i} & =\frac{1}{2}\left({n_{\hat{\theta }}}^{T}\nabla_{q}\psi \right){n_{\hat{\theta }}}^{T}\left(q-q_{i}\right),\;\;\;\forall i\in I_{N} \end{array}$$
respectively, in (Equation 14). Finally, by expressing the aforementioned control laws to the robot’s actual configuration space, we obtain:(21)$$\begin{array}{cc} v & =\nu \hat{v}\\ \omega & =\omega_{dq}+\omega_{d\hat{\theta }} \end{array}$$
where $\omega_{dq}$ and $\omega_{d\hat{\theta }}$ are terms corresponding to angular velocities induced by translational and rotational motion of the robot in the workspace’s image, respectively, given by:$$\begin{array}{cc} \omega_{dq} & =-\hat{v}\nu^{2}{\left({\left(J_{p}\left[\begin{array}{cc} n_{\theta }\; & \nu n_{\theta }^{⊥} \end{array}\right]\right)}^{-1}\frac{\partial }{\partial n_{\theta }}J_{p}n_{\theta }\right)}^{T}\left[\begin{array}{c} 0\\ 1 \end{array}\right]\\ \omega_{d\hat{\theta }} & =\hat{\omega }{\left({\left(J_{p}\left[\begin{array}{cc} n_{\theta }\; & \nu n_{\theta }^{⊥} \end{array}\right]\right)}^{-1}n_{\hat{\theta }}^{⊥}\right)}^{T}\left[\begin{array}{c} 0\\ 1 \end{array}\right] \end{array}$$
with $\nu =∥J_{p}^{-1}n_{\hat{\theta }}∥$ and $\frac{\partial }{\partial n_{\theta }}J_{p}$ denoting the directional derivative of $J_{p}$ along $n_{\theta }$.

The stability properties of the aforementioned closed-loop system dynamics are formalized below.

**Theorem 2.** *The workspace W is invariant under the dynamics of* (Equation 18) *equipped with the proposed control law. Additionally, the robot will asymptotically converge either to an interior critical point of ϕ or to the pre-image of $q_{d}$, which is stable.*

**Proof.** For the proof, refer to the Appendix A.    □

**Remark 3.** 
*The result of Theorem 2 is weaker compared to that of Theorem 1, since there is no guarantee that the set of configurations which converge to a critical point of ϕ (other than the pre-image of $q_{d}$) has Lebesgue measure zero.*


### 5.2. Atlas of Harmonic Maps

As the size of the workspace increases, the problem of computing the transformation *T* grows in complexity as well, because the resources required by commonly employed numerical techniques that can solve the problem presented in Section 3 are polynomial in the number of elements used for representing W. Alternatively, to cope with large workspaces efficiently, we propose instead the construction of an atlas $A≜\{\left(P_{i},T_{i}\right)\;|\;i\in I_{N_{A}}\}$ obtained by separating the workspace W into $N_{A}$ overlapping subsets $P_{i}\subset W$, such that ${⋃}_{i\in I_{N_{A}}}^{}P_{i}=W$ and constructing a separate harmonic map $T_{i}$ for each $P_{i}$ (see Figure 3).

This essentially allows us to solve many small (and computationally less intensive) problems instead of a large one, thus reducing the overall resources required for addressing a given workspace. Therefore, given such a partitioning of W, we define the graph G=(V,E), where $V=\{P_{i}\;|\;i\in I_{N_{A}}\}$ denotes the set of corresponding nodes (workspace partitions) and E denotes the set of edges between the elements of V, with each edge indicating a feasible transition from one partition to another, i.e., (i,j)∈E if and only if $\left(\mathrm{cl}\left(P_{i}\right)\cap \mathrm{cl}\left(P_{j}\right)\right)\neq \emptyset$. Note that G is undirected by definition, i.e., (i,j)∈E only if (j,i)∈E. Additionally, since the workspace is connected, G should also be connected. Thus, for a given atlas A, an initial configuration $p_{\mathrm{init}}$ and a final configuration $p_{d}$, we can employ standard graph search algorithms to obtain a sequence of indices $S=\{s_{1},s_{2},\ldots s_{n}\}$ corresponding to partitions that the robot can tranverse to reach its goal. (In general, more than one such sequence of partitions may exist connecting the initial and the final configurations. However, the selection of one that corresponds to some sort of “optimal” path is beyond the scope of this work.) Additionally, note that since the partitioning of W does not need to be fine, the size of G will generally be small, rendering the cost of finding S negligible.

We now concentrate on how the transition between two consecutive elements of S is implemented. Let $C_{i,j}≜\mathrm{cl}\left(P_{i}\right)\cap \mathrm{cl}\left(P_{j}\right)$ denote the common region of $\mathrm{cl}\left(P_{i}\right)$ and $\mathrm{cl}\left(P_{j}\right)$ and let $B_{i,j}≜\partial P_{i}\cap P_{j}$ denote the set of points on the boundary of $P_{i}$ that also belong to $P_{j}$ and are not occupied by obstacles for all $i\in I_{N_{A}}$ and all *j* such that (i,j)∈E. Without loss of generality, we assume that A is constructed such that the sets $B_{\ell ,i}\cap B_{\ell ,j}$ are either empty or consist of isolated points. We note that in order to successfully complete the transition between two consecutive nodes $P_{i}$ and $P_{j}$ of S, it suffices for the robot to reach any single point of $B_{i,j}$ from $P_{i}$. We also observe that each $B_{i,j}$ may consist of one or more disjoint components $B_{i,j}^{\ell }$, ℓ∈L(i,j), with L(i,j) being some valid indexing of those. By exploiting the fact that Theorem 2 [38] imposes a weak homeomorphism requirement on ${\bar{T}}_{i}$, we can construct each $T_{i}$ such that each disjoint subset of $\partial P_{i}$ collapses into a separate point, i.e., $T_{i}\left(B_{i,j}^{\ell }\right)=q_{i,j}^{\ell }\in \partial D$ (see Figure 3), which, in turn, implies that selecting $q_{i,j}^{\ell }$ as an intermediate goal configuration suffices to render the entire $B_{i,j}^{\ell }$ attractive. Building upon this fact, for each consecutive pair of $P_{i}$ and $P_{j}$ in S, we (arbitrarily) select a $B_{i,j}^{\ell }$ and we construct a transformation $T_{i}:P_{i}\mapsto D$, with $q^{\left[i\right]}=T_{i}\left(p\right)$, and artificial potential field $\phi_{i}\left(q^{\left[i\right]},k^{\left[i\right]}\right)$ with goal configuration $q_{d}^{\left[i\right]}=q_{i,j}^{\ell }$. Additionally, to smooth the transition between consecutive partitions, when they overlap, we propose the following modified control law for the robot:(22)$$u=u^{\left[i\right]}+\eta_{c,i,j}\cdot \eta_{t,i,j}\cdot u^{\left[j\right]},\;\;\;\forall p\in C_{i,j}$$
where $u^{\left[i\right]}$ and $u^{\left[j\right]}$ denote the control inputs as defined in (Equation 6) and evaluated using $\psi_{i},T_{i}$ and $\psi_{j},T_{j}$, respectively; the function $\eta_{t,i,j}:C_{i,j}\mapsto \left[0,1\right]$ is any smooth bump function such that
$$\eta_{t,i,j}\left(p\right)=\left\{\begin{array}{cc} 0, & \;\mathrm{if}\;p\in B_{j,i}\\ 1, & \;\mathrm{if}\;p\in B_{i,j} \end{array}\right.$$
and
$$\eta_{c,i,j}\left(p,k^{\left[i\right]},k^{\left[j\right]}\right)=\left\{\begin{array}{cc} \frac{{\left(\zeta_{i,j}\right)}^{2}}{\epsilon_{4}+{\left(\zeta_{i,j}\right)}^{2}}, & \;\mathrm{if}\;\zeta_{i,j}\geq 0\\ 0, & \;\mathrm{if}\;\zeta_{i,j}<0 \end{array}\right.$$
with $\zeta_{i,j}={\left(\nabla_{p}\psi_{i}\right)}^{T}\cdot \left(\nabla_{p}\psi_{j}\right)$; and $\epsilon_{4}>0$ is a fixed parameter. What this modification essentially does is incrementally add an extra component, with the direction of $\nabla_{p}\psi_{j}$, to the robot’s velocity when that component is cosine similar (two vectors *u* and *v* are cosine similar if their inner product is positive) with $\nabla_{p}\psi_{i}$. We note that $\eta_{c,i,j}\to 1$ and $\eta_{t,i,j}\to 1$ as the robot approaches the boundary of the corresponding partition. We also remark that once the robot has completed its transition to $P_{j}$, we do not concern ourselves with $u^{\left[i\right]}$ anymore, i.e., $u=u^{\left[j\right]}$ even if *p* returns to $C_{i,j}$. The overall scheme employed for navigating a holonomic robot to its goal configuration using an altas constructed as described above can be found in Algorithm 1.

Regarding the stability analysis of the modified system, by following the same procedure as in Section 4.3 and by virtue of $\eta_{c,i,j}$, it is trivial to verify the following statement.

**Theorem 3.** *System* (Equation 1) *equipped with Algorithm 1 converges safely to a given goal configuration $p_{d}\in W$ from almost all initial configurations $p_{\mathrm{init}}\in W$.*

**Proof.** For the proof, refer to the Appendix A.    □

**Algorithm 1** Altas-based motion planning scheme for a holonomic robot**Require:** A, $p_{\mathrm{init}}$, $p_{d}$S←FindPathToGoal($G,p_{\mathrm{init}},p_{d}$)Initialize $k^{\left[s\right]}$ for all s∈S.**for all** *i* in $I_{n-1}$ **do**    $s,s^{\prime }\leftarrow s_{i},s_{i+1}$    Select (arbitrary) *ℓ* such that $\ell \in L(s,s^{\prime })$.    Place goal configuration of $\psi_{s}$ at $q_{s,s^{\prime }}^{\ell }$.**end for**Place goal configuration of $\psi_{s_{n}}$ at $T_{s_{n}}\left(p_{d}\right)$.ℓ←1**loop**    **if** ℓ=n or $p\in P_{s_{\ell }}\backslash P_{s_{\ell +1}}$ **then**        Update *p* using (Equation 6) and $k^{\left[s_{\ell }\right]}$ using (Equation 12).    **else if** $p\in C_{s_{\ell },s_{\ell +1}}$ **then**        Update *p* using (Equation 22) with $i=s_{\ell }$ and $j=s_{\ell +1}$.        Update $k^{\left[s_{\ell }\right]}$ and $k^{\left[s_{\ell }\right]}$ using (Equation 12).    **else**        ℓ←ℓ+1    **end if****end loop**

## 6. Simulations and Experimental Results

In order to demonstrate the efficacy of the proposed control scheme, we have conducted various simulation and experimental studies, the results of which are presented in this section. The algorithm that computes the harmonic transformation and its Jacobian was implemented in C++, while the proposed control protocols were implemented in Python. Code implementations can be accessed at https://github.com/maxchaos/hntf2d (accessed on 16 April 2023). All simulations were carried out on a PC with an Intel i5 processor operating at 2.2 Ghz, with 4 GB RAM and running a GNU/Linux operating system. For more details regarding both simulations and experiments, the reader may refer to the accompanying video material at https://youtu.be/I6WUS81iDh4 (accessed on 16 April 2023).

### 6.1. Simulations—Full Workspace Transformation

In the first case study, a single transformation of the entire 8 m × 5 m workspace (see Figure 3) was constructed and the robot was instructed to navigate to various goal configurations starting from the same initial position. The initial configuration and the parameters of our controller were selected such as to better demonstrate the guaranteed collision avoidance properties of our scheme. Particularly, the initial values for the parameters of the adaptive law were selected as $k_{d}=20$, $k_{i}=1$ and ${\bar{k}}_{i}=20$ for all $i\in I_{10}$. The values of the remaining parameters were $K_{u}=100$, $w_{\phi }=20$, $K_{k}=100$, α=1, $\epsilon_{p}=0.025$, $\epsilon_{v}=0.1$, γ=0.7, $\epsilon_{1}=0.01$, $\epsilon_{2}=0.1$ and $\epsilon_{3}=0.1$. The goal configurations and the trajectories executed by the robot, both in the real and transformed workspace, are illustrated in Figure 4.

The simulations were conducted using the Euler method with 10 ms steps. Regarding the computational complexity of the control scheme, the construction of the harmonic transformation for this large workspace that was carried out offline once required 5.4 s to complete for a sufficient approximation of the workspace boundary with 3680 segments. Finally, the online computation of the transformation T(p) and its Jacobian J(p) required an average of 6.0 ms per step.

### 6.2. Simulations—Atlas of Harmonic Maps

In this case study, we decomposed the aforementioned workspace into separate partitions (see Figure 3) and constructed a harmonic transformation $T_{i}$ for each one (we adopted the door of each room as the common boundary between neighboring partitions). The robot was initialized at the same position as the previous study and it was instructed to navigate towards the same set of individual goal configurations. The initial values selected for the parameters of the adaptive law were $k^{\left[i\right]}=N^{\left[i\right]}+3$, $k_{j}^{\left[i\right]}=1$ and ${\bar{k}}_{j}^{\left[i\right]}=k^{\left[i\right]}$ for all $j\in I_{N^{\left[i\right]}}$ and $i\in I_{N_{A}}$, where $N^{\left[i\right]}$ denotes the amount of obstacles inside the corresponding partition. All remaining control parameters were selected as in Section 6.1. The trajectories of the robot are depicted in Figure 5. The time spent to construct the corresponding harmonic transformations varied from 0.019 s to 0.211 s to (depending on the amount of elements required for sufficiently approximating each room, ranging between 320 and 1000 segments) and was significantly much less than the full map construction of the previous case (5.4 s). Additionally, the online computation of $T_{i}\left(p\right)$ and $J_{i}\left(p\right)$ in each of these rooms required an average time between 1.0 ms and 2.2 ms per step, respectively. Finally, it should be noted that in this case, the workspace inner obstacles were mapped to points further away from the boundaries of the partitions, which is an interesting result as it alleviates possible numerical issues that may arise in the computation of the transformation near the obstacles (the condition number of the Jacobian of the transformation is improved). It should be stressed that the length of the paths in the second case was less (improvement of 0.5 m on average), owing to the fact that the robot gets closer to the workspace boundary since the individual transformations in each room obtain a better conditioned Jacobian (condition number 0.212 against 0.093) and thus are more fine than the first approach, where a transformation is built for the whole workspace.

### 6.3. Comparative Study—Workspace Transformation

In this subsection, we provide a comparative study of the harmonic map presented in this work against readily available workspace transformation methods employed in the motion planning literature. Particularly, we consider four 4 m × 4 m compact workspaces, each associated with a pair of initial and goal positions, and construct appropriate transformations for each one by employing the methodology presented in this work (HM), as well as (i) the Star-to-Sphere Transformation (SST) [15], (ii) the Multi-Agent Navigation Transformation (MANT) [36] and (iii) the Navigation Transformation (NT) [34] (with the aforementioned Star-to-Sphere transformation serving as the underlying map). The trajectories of the robot executed while tracing the line segment connecting the initial and goal configurations in the images of each domain can be seen in Figure 6. We note that manual tuning of the compared transformations was necessary in order to render each a diffeomorphism but without making them too steep around the obstacles. Furthermore, the domain boundaries considered here had to be sufficiently smooth in order for methodologies such as MANT to be applicable. Finally, we remark that the trajectories corresponding to the proposed transformation are, in general, less abrupt compared to the rest, a property attributed to the fact that our approach is global as opposed to the other transformations, i.e., the distortion caused by each obstacle is not limited to some narrow neighborhood around it. The total length, maximum curvature and distance from the obstacles of each executed trajectory can be seen in Table 3, Table 4 and Table 5, respectively. We can see from these values that the actual trajectories yielded using harmonic maps are among the shorter and smoother ones, although they tend to approach the obstacles more than the rest.

### 6.4. Comparative Study—Control Law

In this subsection, we provide a comparative study of our control scheme against other motion planning methodologies.

#### 6.4.1. APF-Based Schemes

To demonstrate the efficacy of the proposed control scheme compared to other APF-based schemes, we considered the 12 m × 16 m workspace depicted in Figure 7, for which we constructed a harmonic map as described in Section 3. Next, we equipped a holonomic robot with three alternative control laws and instructed it to visit four distinct goal positions using these controllers, starting each time from a fixed initial configuration. Particularly, we considered a conventional navigation function-based controller (NF) [15] augmented by [17], for the selection of its notorious parameter, and a harmonic navigation function-based controller (HNF) [33], in addition to our adaptive control scheme (AHNF) described in Section 4. We note that all three control laws considered here make use of the same underlying harmonic map *T* constructed as described above in order to drive the robot to its instructed goal positions. The trajectories executed by the robot can be seen in Figure 7. We remark that, in general, our approach steers the robot away from inner obstacles that lie between its initial and goal configurations, unlike “greedy” schemes such as the conventional NF-based controller, while keeping the traced paths shorter compared to HNFs with fixed source weights, a property attributed to the proposed adaptive laws (Equation 12) which penalize misalignment between the robot’s velocity and the direction towards the goal configuration.

The total length and distance from the obstacles of each executed trajectory can be seen in Table 6 and Table 7, respectively. First, we have to stress that the length trajectory corresponds to the travelled path towards the goal configuration and thus needs to be small, whereas the minimum distance to the workspace boundary refers to the closest point of the trajectory to the workspace boundary and thus needs to be large to have a safe trajectory. Consequently, note from Table 6 that the NF scheme yielded shorter path lengths than the proposed method in two cases (blue and yellow); nevertheless, such paths approach closer to the workspace boundary as indicated in Table 7, thus resulting in more risky paths. On the other hand, the Adaptive Harmonic Potential Field yields a good trade-off between path length and minimum distance to the boundary, since it achieves the shortest paths for two cases without compromising them, as is the case with the NF. On the other hand, the HPF tend to travel around the obstacle closer to the outer workspace boundary and hence exhibit more safe trajectories but they are significantly longer than the other two schemes.

#### 6.4.2. Sampling-Based Scheme

To compare the control scheme proposed in this work against sampling-based methods, we considered a holonomic point-sized robot positioned inside a 6 m × 8 m compact workspace and a desired goal configuration. To complete this task, we employed two different controllers, namely the one proposed in this work and an admissible planner based on an improved probabilistic roadmap method (PRM) [6]. The trajectories executed by the robot using our control law as well as two of the trajectories generated by the PRM-based planner can be seen in Figure 8. The construction of the associated transformation took 31 s to complete for a given boundary approximation made of 7842 elements, whereas the PRM-based planner required approximately 24 s on average over 10 successful runs to yield a solution (we have to stress that we ran 14 trials to get 10 solutions, since four runs did not complete as they exceeded the 500 s calculation time), using the same boundary approximation for collision checking. The robot trajectories exhibited similar lengths in both algorithms (22.5 m for our method against 21.8 m on average), although no path optimization was employed in our case. Additionally, the proposed scheme resulted in a smoother robot trajectory (based on the resulting sequence of points in both cases, we calculated the minimum curvature radius as 0.23 m for our method against 0.12 m on average for the PRM method). On the other hand, note that our approach solves the motion planning problem for any pair of initial and final configurations within the workspace, whereas the sampling-based scheme considers only one go-to problem. Thus, a different initial or final configuration would require a new solution with the PRM method. On the contrary, the proposed transformation needs to be calculated only once to solve the motion planning problem for any pair of initial and final configurations. Finally, it should be noted that for a narrower corridor in Figure 8, the sampling-based approach failed to derive a solution with a reasonable execution time (no solution was calculated within 500 s), since the probability of sampling connected points within this snaky passage reduces drastically. On the contrary, the proposed transformation took 38 s to complete for the same number of elements (i.e., 7842 elements).

### 6.5. Experiments

In order to verify the results presented in Section 5.1, real experiments were conducted on a non-holonomic robotic platform (Robotnik Summit-XL) operating within the 10 m × 25 m compact workspace that is depicted in Figure 9 and Figure 10. The boundaries of the workspace were obtained using readily available SLAM algorithms and were later augmented with the robot’s shape (approximated by a disk). The workspace was partitioned into six overlapping subsets and the robot was instructed to visit three different goal configurations, each located in a different room. An off-the-shelf localization algorithm was employed for estimating the robot’s position and orientation using its on-board sensors (laser scanners and RBG-D cameras), providing feedback at approximately 5 HZ to the robot’s linear and angular commanded velocities. The construction of the associated transformations over the six subsets of the workspace took from 1.3 s for the simple and smaller partitions with 800 elements to 3.1 s for the more complex ones employing 1500 elements. On the other hand, the evaluation of the mapping as well as its Jacobian took less than 6 ms on average, which was satisfactory given the low position update rate. Note that our algorithm successfully managed to drive the robot safely (the minimum distance to the workspace boundary was 0.15 m when passing through the doors) to its specified goal configurations, as one can verify from the trajectories (see Figure 9, Figure 11 and the accompanying video material). However, an issue that needs to be pointed out is the oscillating behavior that the robot exhibited in the configuration space’s image, particularly in subsets p1 and p2 as depicted in Figure 11. Such behavior is attributed both to (a) the relative slow update of the robot’s pose estimation and (b) the inversion of the Jacobian which is ill-conditioned close to extremely narrow passages of the domain. Nevertheless, such shortcomings can be alleviated by a better choice of partitions, e.g., by partitioning the domain into more subsets with less complex shapes. As a future research direction, we shall investigate whether the condition number of the Jacobian of the transformation is a fine criterion, since the condition number is usually used to measure how sensitive a function is to changes or errors in the input, and the output error results from an error in the input via the Jacobian.

## 7. Conclusions and Future Work

In this work, we employed harmonic map theory to devise a transformation of complex workspaces directly to point worlds that are appropriate for robot navigation. Subsequently, we presented a novel motion planning control scheme based on closed-form harmonic potential fields equipped with appropriate adaptive laws for their parameters, which can safely navigate a robot to its goal state from almost all initial configurations. Additionally, we extended our approach to accommodate the navigation problem of non-holonomic robots and kept the numeric computations tractable for large workspaces.

Regarding future directions, our aim is first to increase the applicability of the proposed navigation framework by addressing partially known dynamic workspaces, which is far from being straightforward. To remedy the issue of calculation time in this case, we shall adopt a sensitivity analysis approach so that we do not solve the whole problem from scratch, but find how the solution deviates when a small change in the workspace occurs. In this way, we envision a reasonable calculation time (except from the first calculation) that would result in an almost real-time calculation of the transformation and thus allow us to consider even moving obstacles in dynamic environments. However, critical issues have to be studied concerning cases where the workspace changes topologically (e.g., in the case of antagonistically moving obstacles) and this results in significant changes in the transformation. In the same vein, switching in the transformation output might raise practical issues such as chattering that have to be carefully considered. Note that the aforementioned research direction could also serve as a first step towards the solution of the multi-robot motion planning problem, where for each robot all other robots should be considered as moving obstacles, operating antagonistically to achieve their goal configurations. Finally, another challenging research direction concerns the extension to 3D workspaces. Unfortunately, the harmonic maps have been studied only for 2D workspaces, since they rely heavily on complex analyses. Nevertheless, we propose to decompose the 3D motion planning problem into several 2D sub-problems, where the proposed solution works, and then combine them (e.g., decompose the motion along the *z*-axis and on the *x*-*y* plane).

## Figures and Tables

**Figure 1 sensors-23-04464-f001:**
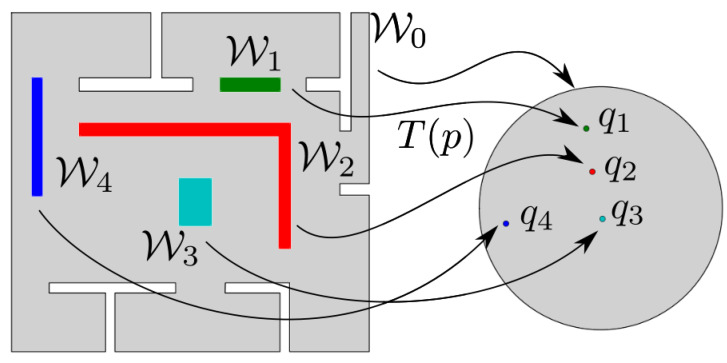
Transformation of a workspace onto a punctured disk.

**Figure 2 sensors-23-04464-f002:**
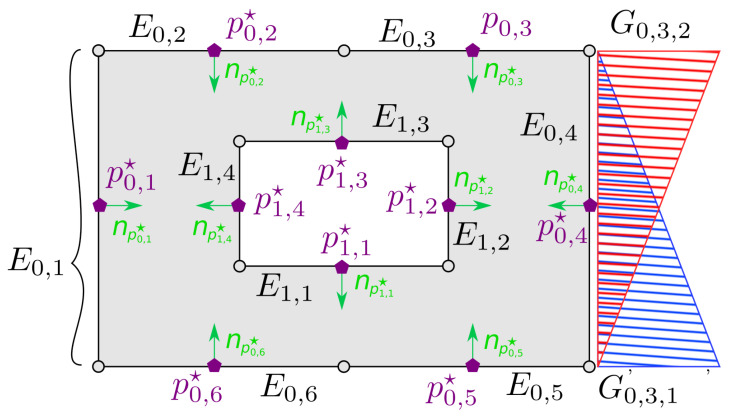
Discretization of a given domain’s boundary using line segment elements. By convention, the outer boundary is considered to be clockwise oriented, whereas inner boundaries are counter-clockwise oriented. The normal direction of each element is depicted using green colored vectors. Furthermore, the values of the two linear shape functions ${\tilde{G}}_{0,3,1}$ and ${\tilde{G}}_{0,3,2}$ are plotted along the associated element ${\tilde{E}}_{0,3}$.

**Figure 3 sensors-23-04464-f003:**
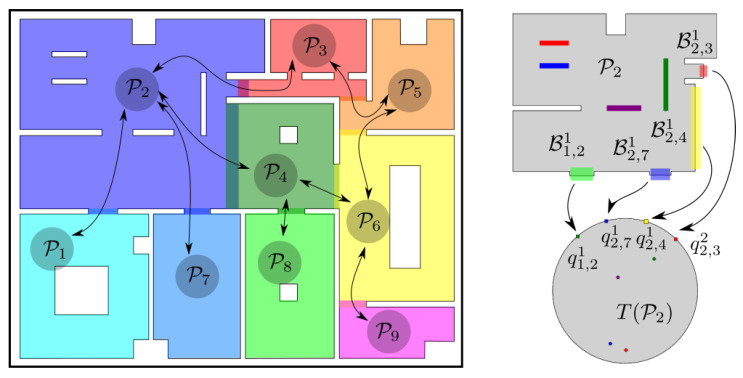
The partition of a complex workspace into overlapping subsets along with the corresponding graph and the tranformation $T_{2}$ of the second partition $P_{2}$.

**Figure 4 sensors-23-04464-f004:**
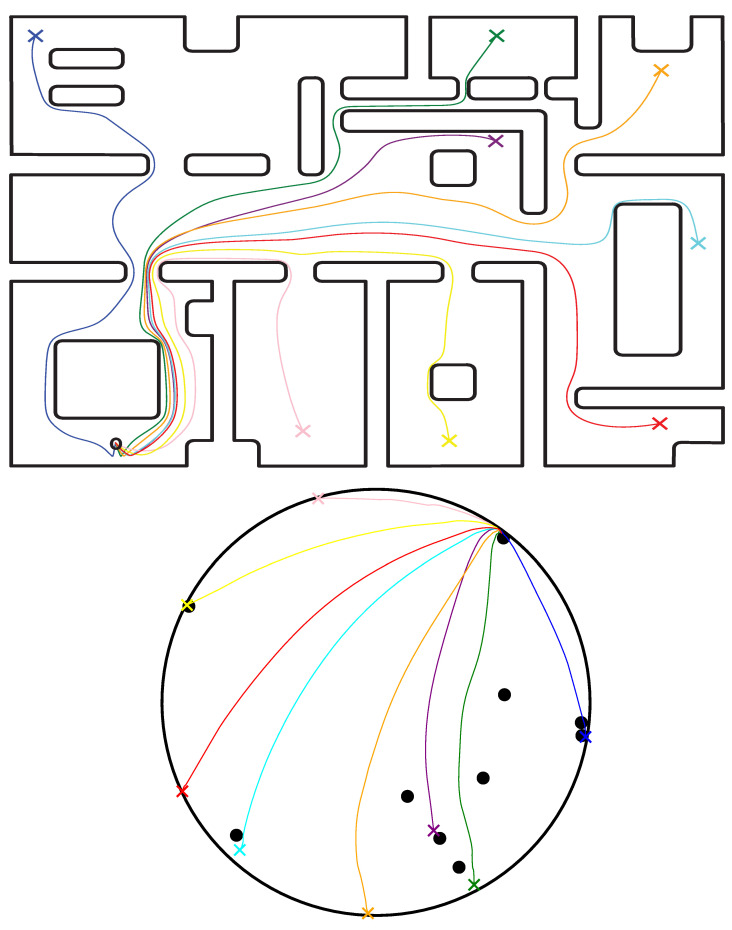
Robot trajectories illustrated with various colors in both the actual workspace (**upper plot**) and their image (**bottom plot**) using a full workspace transformation. The robot starts from an arbitrary location at the bottom left part of the workspace (black circle) and navigates to different goal configurations (colored crosses). Note that the ten black dots in the lower plot correspond to the points where the internal obstacles of the actual workspace have been transformed.

**Figure 5 sensors-23-04464-f005:**
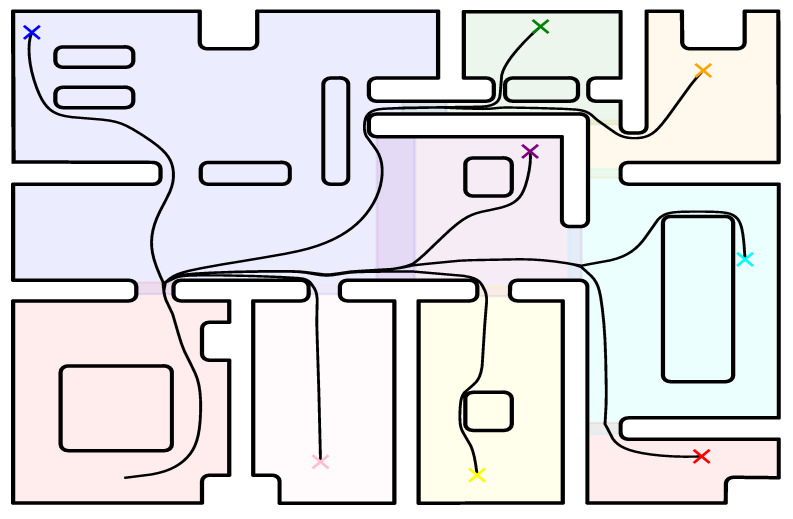
The resulting robot trajectories for various goal configurations depicted with colored crosses, using an atlas of the workspace.

**Figure 6 sensors-23-04464-f006:**
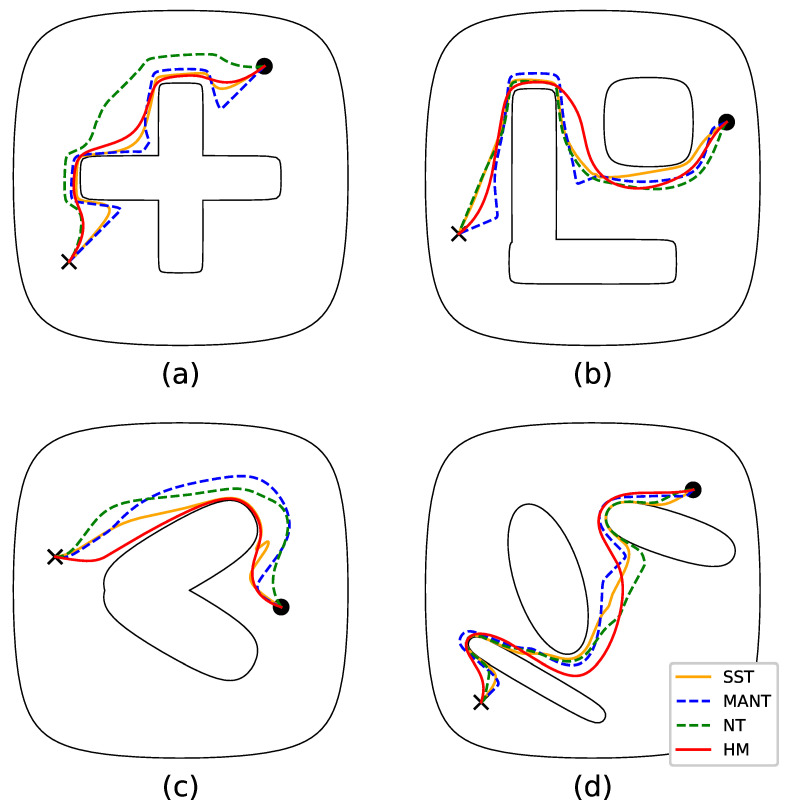
Robot trajectories from initial configuration (black circle) straight to the desired configuration (black cross) by employing various domain transformation methods in each workspace (**a**–**d**).

**Figure 7 sensors-23-04464-f007:**
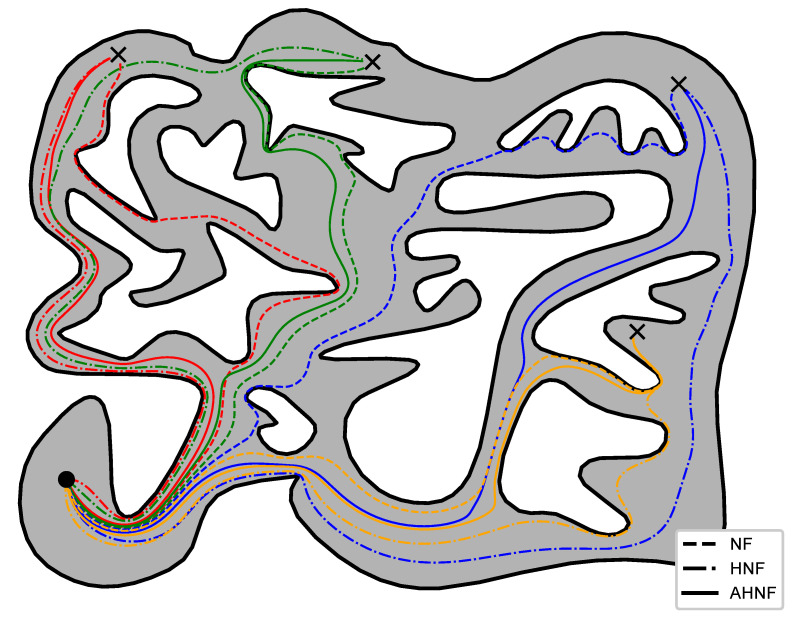
Trajectories of the robot navigating to four distinct goal configurations (black crosses) with red, green, yellow and blue color starting from the same initial position (black circle) while using various alternative APF-based controllers.

**Figure 8 sensors-23-04464-f008:**
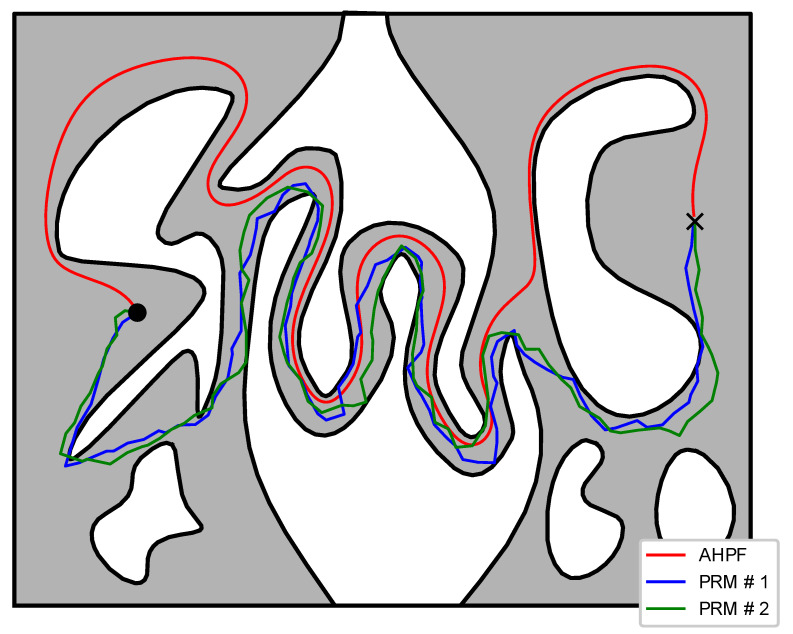
Trajectories of the robot navigating to its goal configuration (black cross) generated using the proposed control law and a PRM-based planner.

**Figure 9 sensors-23-04464-f009:**
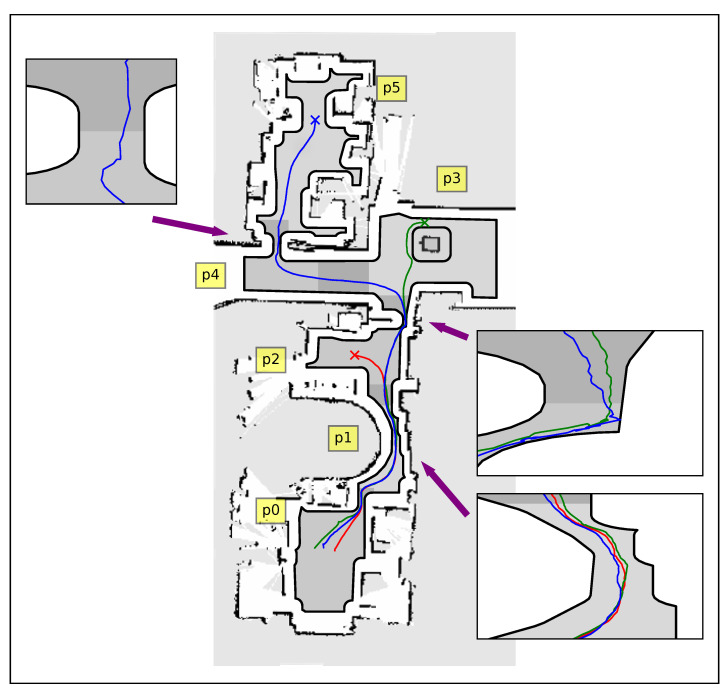
Trajectories of the unicycle-like robot in the real workspace, with each color (red, green and blue) corresponding to a different goal configuration. Dark gray regions indicate areas where partitions overlap.

**Figure 10 sensors-23-04464-f010:**
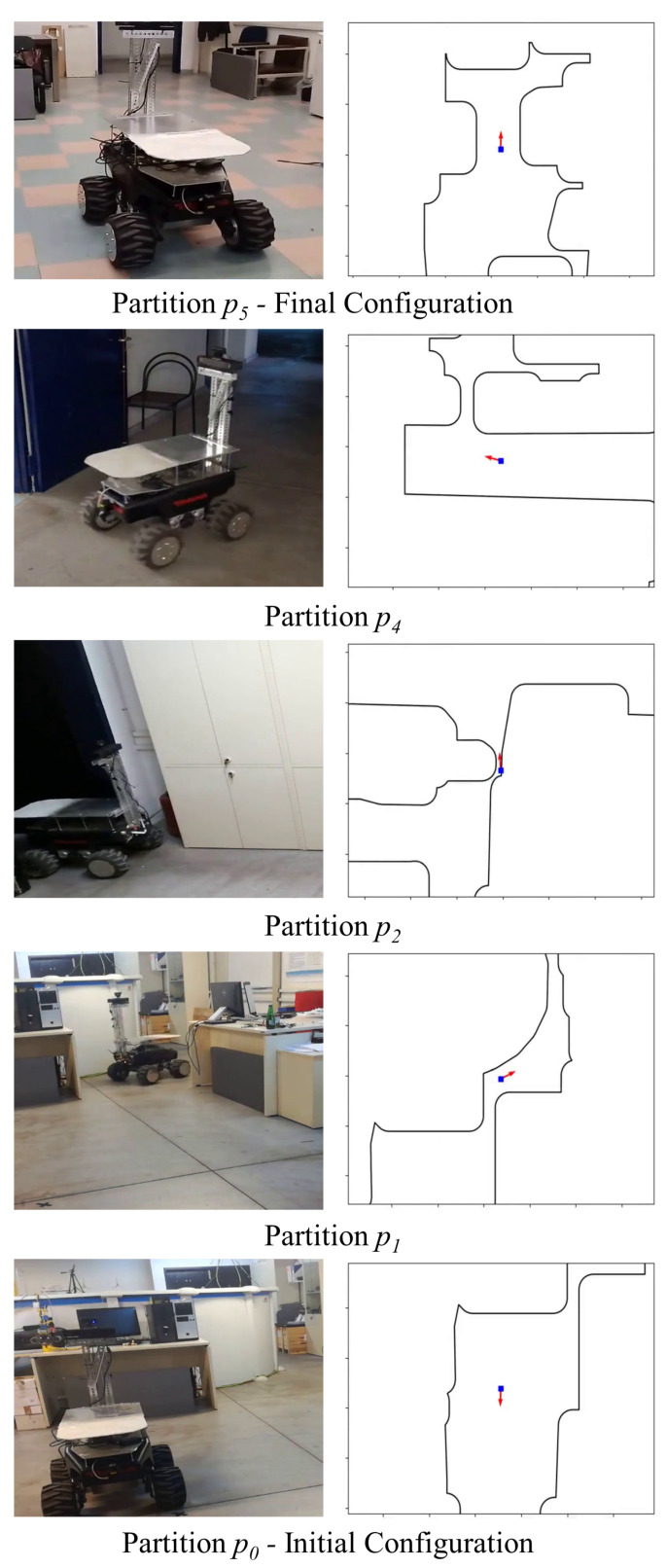
Snapshots of the robot navigating through the workspace. The figures on the right illustrate the position and orientation of the robot with respect to the workspace map.

**Figure 11 sensors-23-04464-f011:**
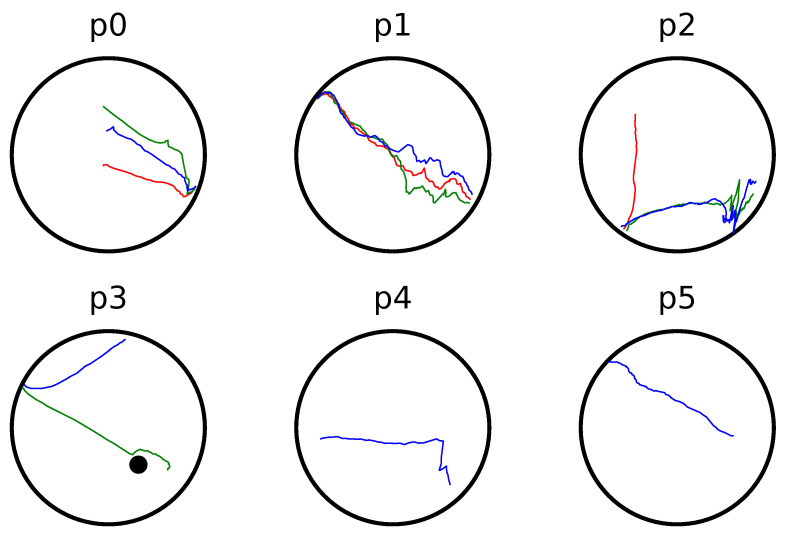
Robot image trajectories within each partition of the workspace for three experiments in red, green and blue color.

**Table 1 sensors-23-04464-t001:** Comparison between the Harmonic Transformation (HM) proposed in this work and the (i) Star-to-Sphere Transformation (SST) [15], (ii) Multi-Agent Navigation Transformation (MANT) [36] and (iii) the Navigation Transformation (NT) [34]. Although HMs require global knowledge of the workspace’s geometry to be constructed, HMs are infinitely differentiable and require the domain to be represented by closed polygonal curves (which can be easily obtained using SLAM methodologies), unlike the alternatives that require the domain boundaries to be represented as sets of sufficiently differentiable implicit equations.

	Geometry Representation	Global	Analytic
HM	Points on the boundary	Yes	Yes
SST	Trees of Stars	Yes	Yes
NT	$C^{2}$-manifolds	No	No
MANT	Trees of $C^{2}$-manifolds	No	No

**Table 2 sensors-23-04464-t002:** Comparison of Adaptive Harmonic Potential Fields (proposed herein) with common alternatives, specifically Rimon–Koditchek Navigation Functions (RKNF) [15], Harmonic Navigation Functions [33] and approximate Harmonic Potential Fields obtained using numerical techniques [21]. Unlike RKNFs that require tuning for ensuring convergence to the goal from almost all initial configurations and HNFs that require tuning for guaranteeing collision avoidance with the workspace boundaries, the proposed control law enjoys both properties by design.

	Convergence	Collision Avoidance	Computational Cost
AHPF	By design	By design	Cheap
HPF	By design	By design	Expensive
RKNF	Requires tuning	By design	Cheap
HNF	By design	Requires tuning	Cheap

**Table 3 sensors-23-04464-t003:** Trajectory lengths (m) executed by employing the four alternative transformations in each workspace displayed in Figure 6.

	a	b	c	d
SST	3.63	4.18	2.12	4.09
MANT	4.26	4.69	2.34	4.45
NT	3.35	4.30	2.18	4.22
HM	3.19	4.21	2.05	4.32

**Table 4 sensors-23-04464-t004:** Maximum value of curvature (m^−1^) associated with each trajectory displayed in Figure 6.

	a	b	c	d
SST	4.22	5.43	86.93	2.16
MANT	1.23	1.47	13.56	2.06
NT	66.97	25.23	14.89	6.92
HM	2.47	2.49	14.76	2.77

**Table 5 sensors-23-04464-t005:** Minimum distance (m) between each robot trajectory and the corresponding workspace boundaries displayed on Figure 6.

	a	b	c	d
SST	0.0303	0.0283	0.0159	0.0063
MANT	0.0644	0.1253	0.1870	0.0648
NT	0.1386	0.0506	0.0915	0.0058
HM	0.0335	0.0377	0.0103	0.0181

**Table 6 sensors-23-04464-t006:** Length of trajectories (m) executed by each controller (Rimon–Koditchek Navigation Functions, Harmonic Navigation Functions and Adaptive Harmonic Potential Fields) in Figure 7.

	Red	Green	Blue	Yellow
NF	19.781	20.427	**22.090**	**18.397**
HNF	18.224	22.538	26.959	20.062
AHNF	**17.874**	**19.419**	23.364	18.595

**Table 7 sensors-23-04464-t007:** Minimum distance (m) between the corresponding workspace boundaries and each trajectory displayed in Figure 7.

	Red	Green	Blue	Yellow
NF	0.1158	0.0102	0.1210	0.1103
HNF	**0.3347**	**0.2135**	**0.2591**	**0.2166**
AHPF	0.1310	0.0352	0.2043	0.1854

## Data Availability

Not applicable.

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
