# Peer review of "Robot Navigation in Complex Workspaces Employing Harmonic Maps and Adaptive Artificial Potential Fields"

_sensors, 2023, doi:10.3390/s23094464_

Round 1

Reviewer 1 Report

General Comment

This article proposes a path planning method that combines harmonic maps and adaptive artificial potential fields to solve navigation problems for wheeled robots in planar and arbitrarily connected workspaces. However, there are some issues that need to be addressed before publication.

Specific Comments:

1. The methodology presented in this article contains sections that are overly verbose. The explanation of artificial potential fields, in particular, is already a well-established technology, and some content can be condensed to highlight the innovation of the proposed method.

2. While the experiments in the article demonstrate various scenarios, it would be helpful to have additional data, such as the smoothness and efficiency of the generated paths, to verify the performance of the proposed method.

3. It is recommended that the author review the labeling conventions used throughout the article for scalars, variables, and vectors in the equations to ensure consistency and accuracy

The suggestion is for the author to further improve the paper in terms of language and grammar. For example, in some cases, the wording can be simplified to improve readability, and in some cases, subject-verb agreement can be improved. In addition, some equations and technical terms could benefit from further clarification.

Author Response

Comments and Suggestions for Authors

General Comment

This article proposes a path planning method that combines harmonic maps and adaptive artificial potential fields to solve navigation problems for wheeled robots in planar and arbitrarily connected workspaces. However, there are some issues that need to be addressed before publication.

Specific Comments:

Q1. The methodology presented in this article contains sections that are overly verbose. The explanation of artificial potential fields, in particular, is already a well-established technology, and some content can be condensed to highlight the innovation of the proposed method.

A1. Thanks for the comment. We omitted from the first paragraph of the introduction of the original manuscript the information on artificial potential fields that is well established in the related literature. Towards this direction, we reformed the introductory section in such way that the related works are compared with the proposed method to put forward more clearly the contribution of our work.

Q2. While the experiments in the article demonstrate various scenarios, it would be helpful to have additional data, such as the smoothness and efficiency of the generated paths, to verify the performance of the proposed method.

A2. Thanks for the comment. Regarding the experiment with the real robot in Section 6.5, we have to stress that we only illustrate the trajectories since the goal of the experimental study is to verify that the proposed transformation and control scheme can be implemented in real-time on an actual robotic system with limitations in the sensing. Performance metrics were only considered during the comparative simulation studies as implementing methodologies on a real robot that have been presented in other papers is beyond the scope of this paper. Nonetheless, a smoothness metric (minimum curvature radius) was adopted in the simulated comparison of the proposed method with a sampling-based method (PRM) in Subsection 6.4.2 (refer to the revised manuscript). Other metrics such as path length and minimum distance to the workspace boundary have also been considered in Subsection 6.4.1 where other potential fields methods are compared in simulation (Tables 6 and 7). Similar metrics have been considered when comparing in simulation the proposed transformation with others in Subsection 6.3 (Tables 3, 4 and 5). Regarding optimality/efficiency, since our approach is reactive it decides where to direct based on its current position, hence optimality has not been considered as for optimality you have to look ahead in order to select the shortest path. Nevertheless, we are currently working on optimal motion planning based on Reinforcement learning to improve the input energy and the path length.

Q3. It is recommended that the author review the labeling conventions used throughout the article for scalars, variables, and vectors in the equations to ensure consistency and accuracy

A3. The authors put their best efforts to correct any typos in the equations within the revised manuscript.

Comments on the Quality of English Language

Q. The suggestion is for the author to further improve the paper in terms of language and grammar. For example, in some cases, the wording can be simplified to improve readability, and in some cases, subject-verb agreement can be improved. In addition, some equations and technical terms could benefit from further clarification.

A. Thanks for the comment. The authors put their best efforts: a) to correct any typos and syntax discrepancies within the revised manuscript, as well as b) to enhance the presentation.

Reviewer 2 Report

1. While the authors briefly mentioned potential extensions to 3D workspaces and multi-robot systems, it would be helpful to append a more detailed discussion on the limitations of the proposed approach and potential areas for future research. This would give readers a more comprehensive understanding of the current state of the work and its potential for future development.

2. In section 6.4.2. of Sampling-based Scheme, it is necessary to append more specific discussion on the outperformance of smoother robot trajectory. Also, the compared method PRM in the reference [6] was performed on their method not so much sharp trajectory as represented in this paper. Furthermore, is there any special reason for not using of path optimization?

3. In the line 392-397, the paragraph represents the discussion on Table 6 and 7. In the Table 6, it seems that AHNF has best performance for the length of trajectory by each controller. However, in the Table 7, it seems taht NF seems to be the best for minimum distances. It is required for the discussion of the unique results by the methodology differences.

4. Figure 4 does not show what each tractor represents. It seems that the figure needs to be modified.

5. In line 394, It is mentioned that NF is the shortest, but it's the longest in table 6 Red. Is the simulation result reasonable?

6. In the appendix, if all three terms from which V is differentiated are non-positive, then the V function needs to be described as positive to satisfy the Lyapunov function and prove system stability, and It is required to add that the expression V is positive when defining V.

Some expressions needs to be corrected in grammatical way.

Author Response

Comments and Suggestions for Authors

Q1. While the authors briefly mentioned potential extensions to 3D workspaces and multi-robot systems, it would be helpful to append a more detailed discussion on the limitations of the proposed approach and potential areas for future research. This would give readers a more comprehensive understanding of the current state of the work and its potential for future development.

A1. Thanks for the comment. We omitted the corresponding sentence from the abstract as it is unrealistic to present there a detailed future work plan. Nevertheless, we revised accordingly the corresponding part in the concluding section including limitations and future plans (Section 7). More specifically, we mention that:

“Regarding future directions, our aim is first to increase the applicability of the proposed navigation framework by addressing partially known dynamic workspaces, which is far from being straightforward. To remedy the issue of calculation time in this case, we shall embark into a sensitivity analysis approach so that we don’t solve the whole problem from scratch, but find how the solution deviates when a small change in the workspace occurs. In this way, we envision a reasonable calculation time (except from the first calculation) that would result in an almost real-time calculation of the transformation and thus allow us to consider even moving obstacles in dynamic environments. However, critical issues have to be studied concerning cases where the workspace changes topologically (e.g., in case of antagonistically moving obstacles) and this results in significant changes in the transformation. In the same vein, switching in the transformation output might raise practical issues like chattering that has to be carefully considered. Notice that the aforementioned research direction could also serve as a first step towards the solution of the multi-robot motion planning problem, where for each robot all other robots should be considered as moving obstacles, operating antagonistically to achieve their goal configurations. Finally, another challenging research direction concerns the extension to 3D workspaces. Unfortunately, the harmonic maps have been studied only for 2D workspaces, since they rely heavily on complex analysis. Nevertheless, we propose to decompose the 3D motion planning problem into several 2D sub-problems, where the proposed solution works and then combine them (e.g., decompose the motion along the $z$-axis and on the $x$-$y$ plane).”

Q2. In section 6.4.2. of Sampling-based Scheme, it is necessary to append more specific discussion on the outperformance of smoother robot trajectory. Also, the compared method PRM in the reference [6] was performed on their method not so much sharp trajectory as represented in this paper. Furthermore, is there any special reason for not using of path optimization?

A2. In Figure 8, the illustrated paths resulted by the PRM technique correspond to the points that are calculated by PRM (each point can be connected to its previous and next one by line segments owing to the visibility constraint incorporated in PRM). Apparently, we plot the points and connect the neighboring ones in the figure since otherwise just illustrating the points would not be very helpful. The results presented in reference [6] have been obtained by smoothly interpolating the resulted points by the PRM method. Based on the resulted sequence of points in both cases, we calculated the minimum curvature radius following the same numerical rule and obtained smoother paths with the proposed method. Regarding optimality, since our approach is reactive it decides where to direct based on its current position, hence optimality has not been considered since for optimality you have to look ahead in order to select the shortest path. Nevertheless, we are currently working on optimal motion planning based on Reinforcement learning to improve the input energy and the path length.

Q3. In the line 392-397, the paragraph represents the discussion on Table 6 and 7. In the Table 6, it seems that AHNF has best performance for the length of trajectory by each controller. However, in the Table 7, it seems that NF seems to be the best for minimum distances. It is required for the discussion of the unique results by the methodology differences.

A3. First, we have to stress that the length trajectory corresponds to the travelled path towards the desired position and thus needs to be small, whereas the minimum distance to the workspace boundary refers to the closest point of the trajectory to the workspace boundary and thus needs to be large to have a safe trajectory. Consequently, notice from Table 6 that NF yielded shorter path lengths than the proposed method in two cases (blue, yellow); nevertheless, such paths approach closer to the workspace boundary as indicated in Table 7, thus resulting in more risky paths. On the other hand, the Adaptive Harmonic Potential Field yields a good trade-off between path length and minimum distance to the boundary since it achieves the shortest paths for two cases without compromising them as done by the NF. On the other hand, the HPF as they tend to travel around the obstacle closer to the outer workspace boundary exhibiting more safe trajectories but significantly longer than the other two schemes. The corresponding part of the manuscript has been revised to clarify these issues.

Q4. Figure 4 does not show what each tractor represents. It seems that the figure needs to be modified.

A4. In the upper plot of Figure 4, we illustrate the yielded robot trajectories starting from the initial configuration (black circle at the bottom left) towards various goal configurations in the workspace (colored crosses). The bottom plot illustrates the image of the trajectories on the transformed workspace (punctured disk). Notice that the dots correspond to the points where the internal obstacles have been transformed similarly to Figure 3. The caption of Figure 4 has been revised to clarify both plots.

Q5. In line 394, It is mentioned that NF is the shortest, but it's the longest in table 6 Red. Is the simulation result reasonable?

A5. As we mention in your previous related comment the NF-approach yielded shorter path lengths than the proposed method in two cases (blue, yellow). This is attributed mainly to the fact that NFs are “greedy”, so if not so many curved/complex obstacles are met then the robot moves exactly towards the goal configuration. On the other hand, in red and blue cases the NF-approach tends to closely trace the inner obstacles boundaries thus resulting in longer trajectories.

Q6. In the appendix, if all three terms from which V is differentiated are non-positive, then the V function needs to be described as positive to satisfy the Lyapunov function and prove system stability, and It is required to add that the expression V is positive when defining V.

A6. Notice that the candidate Lyapunov function V is selected equal to the reference potential $\psi$ in (10), which is positive and takes values within the interval [0,1] (the minimum value 0 is attained at the desired configuration q=q_d as clearly indicated by (9)). Differentiating the aforementioned Lyapunov candidate function along the system’s trajectories as given in (28), we obtain a non-positive time derivative which concludes stability and based on LaSalle’s invariance theorem convergence to the largest invariant set of the system. We have revised accordingly the equations to make the presentation clearer.

Comments on the Quality of English Language

Q. Some expressions need to be corrected in grammatical way.

A. The authors put their best efforts to correct any typos and syntax discrepancies within the revised manuscript.